# Fourier Token Merging: Understanding and Capitalizing Frequency Domain for Efficient Image Generation

**Jiesong Liu**
Department of Computer Science
North Carolina State University
Raleigh, NC 27606
jliu93@ncsu.edu

**Xipeng (Gracen) Shen**
Department of Computer Science
North Carolina State University
Raleigh, NC 27606
xshen5@ncsu.edu

## Abstract

Image generation requires intensive computations and faces challenges due to long latency. Exploiting redundancy in the input images and intermediate representations throughout the neural network pipeline is an effective way to accelerate image generation. Token merging (ToMe) exploits similarities among input tokens by clustering them and merges similar tokens into one, thus significantly reducing the number of tokens that are fed into the transformer block. This work introduces *Fourier Token Merging*, a new method for understanding and capitalizing frequency domain for efficient image generation. By introducing frequency token merging, we find that transforming the token into the frequency domain representation for clustering can better exert the ability of clustering based on the underlying redundancy after de-correlation. Through analytical and empirical studies, we demonstrate the benefits of using Fourier clustering over the original time domain clustering. We experimented *Fourier Token Merging* on the stable diffusion model, and the results show up to 25% reduction in latency without impairing image quality. The code is available at `https://github.com/Fred1031/Fourier-Token-Merging`.

## 1 Introduction

Image generation has achieved impressive results through advanced generative models such as stable diffusion [21], but these models face significant computational challenges and long latency, especially when deployed on resource-constrained devices [5, 11, 26]. A primary avenue for addressing these issues involves exploiting redundancies within input images and intermediate neural network representations to streamline computation. *Token merging* (ToMe) represents one such approach, utilizing clustering methods to merge similar tokens, thus substantially reducing computational overhead within transformer blocks [2].

In this work, we introduce *Fourier Token Merging*, a novel technique that leverages frequency-domain insights for efficient clustering and merging of tokens during image generation. Unlike traditional methods, *Fourier Token Merging* transforms tokens into frequency-domain representations, capitalizing on the inherent redundancy revealed after decorrelation. By clustering tokens in this frequency space, our method enhances clustering effectiveness and significantly reduces computational latency.

We present both theoretical and empirical analyses demonstrating that frequency-domain clustering surpasses conventional time-domain clustering approaches in capturing token similarities. Specifically, we show analytically that Fourier-based clustering reduces token error within clusters, especially as the diffusion process progresses and as tokens propagate into deeper transformer layers. Empirical results from extensive experiments on the stable diffusion v1.5 model validate our approach, achieving up to a 25% latency reduction without sacrificing image quality.

39th Conference on Neural Information Processing Systems (NeurIPS 2025).

To the best of our knowledge, this is the first work to explore frequency domain for speeding up image generation. Our results underline the potential of frequency-domain techniques to enhance the efficiency and practicality of advanced generative models, opening new possibilities for deploying high-quality image generation on edge devices and low-resource environments.

## 2 Background

**Token Merging for Image Generation.** *Token merging* [2, 3] has emerged as an effective technique to optimize computational efficiency in generative image models such as stable diffusion. By grouping similar tokens and merging them, these methods substantially reduce computational overhead and latency during image generation processes. While most token merging methods focus on standard Vision Transformers, Token Merging with Attention (ToMA) [14] is a key prior work that first adapted token merging for DiT-like models (e.g., SDXL and Flux) by introducing a strategy to handle their specific architectural challenges. More related work to reduce transformer computaion is in Appendix C.

**Discrete Fourier Transform.** Central to our approach is the Fast Fourier Transform (FFT), a widely used computational algorithm that efficiently computes the Discrete Fourier Transform (DFT). The DFT decomposes a spatiotemporal signal into frequency-domain representations, revealing correlations at different scales and orientations. For a given input $\mathbf{x} = \{x_0, \cdots, x_{N-1}\}$, the DFT is defined as:

$$\mathcal{F}(\mathbf{x})_k = \sum_{n=0}^{N-1} x_n e^{-\frac{2\pi i}{N} nk}, 0 \leq k \leq N - 1. \tag{1}$$

The inverse transform, which reconstructs the original signal from its frequency-domain representation, is simply the conjugate transpose of the forward transform, emphasizing the linearity and unitarity of the DFT. The FFT algorithm [6, 8] significantly reduces computational complexity from $O(N^2)$ to $O(N \log N)$, facilitating rapid signal processing and making it practical for real-world applications. Its computational efficiency and versatility have made it a standard tool across various domains, notably signal and image processing. This background underpins our novel *Fourier Token Merging* technique, leveraging frequency-domain insights to enhance image generation efficiency.

## 3 Fourier Token Merging for Image Generation

To accelerate inference in diffusion models on the fly without compromising visual fidelity, we introduce a novel merging framework that leverages frequency-domain representations. Our approach, termed *Fourier Token Merging*, augments existing token reduction techniques by incorporating global structural priors via the Discrete Fourier Transform (DFT). Figure 1 illustrates the architecture of the proposed system.

### 3.1 Modular Integration in the Diffusion Pipeline

Our framework is integrated into the denoising backbone of a diffusion model, such as the U-Net used in Stable Diffusion. At each sampling timestep $t$, the model receives a noisy latent $z_t$ and computes its denoised counterpart through a modified $\epsilon_\theta$ network that supports frequency-domain token merging.

The Original Merging Module represents baseline spatial-domain merging. The data are first clustered in a token-level granularity and then each cluster uses the centroid to represent the whole cluster and uses only that one token for further transformer block computation. The bipartite softmatch clustering component partitions the tokens into two sets (`src` and `dst`) and computes the similarity score between each of the `src` and `dst` pair. For each token in the `src`, the module clusters it with the corresponding token in the `dst` with the highest similarity score. The module then computes attention operations with the reduced tokens. Since the subsequent ResNet requires the full number of tokens, tokens are unmerged to their original length, optimizing the computation speed in the attention module. ResNet blocks and MLP layers remain unchanged.

The Fourier Merging Module transforms each token $\mathbf{x} \in \mathbb{R}^d$ into the frequency domain via the Discrete Fourier Transform (DFT), denoted by $\mathcal{F}(\mathbf{x})$ as introduced in Section 2. To reduce noises

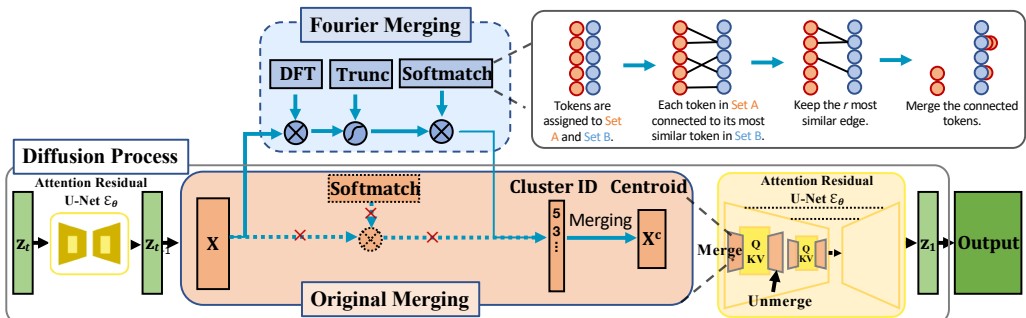

Figure 1: The proposed *Fourier Token Merging* Framework and the original *Token Merging* for Diffusion Models. *Fourier Token Merging* integrates a frequency-aware merging mechanism into the diffusion generation process. The Original Merging Module from the traditional method performs token merging based on similarity in the spatial domain. In contrast, the *Fourier Merging Module* processes intermediate visual representations using the Discrete Fourier Transform before truncating the higher frequencies, enabling structured token merging in the frequency domain using a clustering method called *softmatch*. The tokens are merged into the *Attention Residual U-Net* ($\epsilon_\theta$), which computes the denoised prediction at each timestep. The figure shows the Diffusion Pipeline, where the model iteratively denoises latent variables from $z_t$ down to the final output. This modular design allows Fourier-based merging in place of existing merging techniques, improving both generation efficiency and preservation of global structure.

while preserving structural semantics, we retain only the dominant low-frequency components and discard the rest. This is done by truncating the real part of the DFT output according to a threshold $\tau \in (0, 1]$:

$$\tilde{\mathbf{x}} = \text{Trunc}_\tau \left( \text{Re} \left( \mathcal{F}(\mathbf{x}) \right) \right) \tag{2}$$

Here, $\mathcal{F}(\mathbf{x}) \in \mathbb{C}^d$ is the full complex-valued spectrum, $\text{Re}(\cdot)$ extracts the real-valued coefficients, and $\text{Trunc}_\tau(\cdot)$ retains only the lowest $\tau d$ frequency components. The resulting representation $\tilde{\mathbf{x}}$ is then used for similarity computation and token clustering, effectively emphasizing coarse-grained structure while suppressing high-frequency noise.

The clustering component employing the same bipartite soft matching then identifies and clusters tokens based on the frequency structures. This improves clustering according to the underlying structures of the tokens after decorrelation of the time domain information. The clustering results are then used to guide the merging process, according to which the original tokens are merged. The reduced token set is then forwarded to the self-attention and cross-attention layers within the U-Net.

Once merged, the modified U-Net computes the denoised residual $\epsilon_\theta(z_t, t)$, which is used in a standard reverse diffusion step to obtain $z_{t-1}$. This process is repeated iteratively until the final clean latent $z_0$ is reached. By performing *Fourier Token Merging* at every timestep, the model enjoys consistent computational savings throughout the generation trajectory.

In contrast to purely spatial methods, *Fourier Token Merging* enables:

(1) **underlying redundancy detection:** Frequency representations capture long-range dependencies compactly, aiding the retention of layout and structure. The module exerts the ability to cluster based on the underlying redundancy after decorrelation.

(2) **compact representation:** Many high-resolution regions exhibit sparsity in the frequency domain, allowing more aggressive merging without degradation.

(3) **compatibility:** The module is lightweight, plug-and-play, and can be applied to pretrained diffusion models without retraining.

### 3.2 Theoretical Approximation Error Analysis

We now analyze the approximation error introduced by Token Merging as a preparation for analyzing the benefits of *Fourier Token Merging*. For a given input sequence $\mathbf{X} := [\mathbf{x}_1, \mathbf{x}_2, \cdots, \mathbf{x}_N]^\top \in \mathbb{R}^{N \times D_x}$ of $N$ feature vectors with bounded norm $\|\mathbf{x}_j\| \leq R$, $j = 1, \cdots, N$, self-attention transforms $\mathbf{X}$ into the output sequence $\mathbf{H}$ in the following two steps:

1. The input sequence $\mathbf{X}$ is projected into the query matrix $\mathbf{Q}$, the key matrix $\mathbf{K}$, and the value matrix $\mathbf{V}$ via three linear transformations

$$\mathbf{Q} = \mathbf{W}_Q \mathbf{X}; \quad \mathbf{K} = \mathbf{W}_K \mathbf{X}; \quad \mathbf{V} = \mathbf{W}_V \mathbf{X},$$

where $\mathbf{W}_Q, \mathbf{W}_K \in \mathbb{R}^{D \times D_x}$, and $\mathbf{W}_V \in \mathbb{R}^{D_v \times D_x}$ are the weight matrices. Let $\mathbf{Q} := [\mathbf{q}_1, \cdots, \mathbf{q}_N], \mathbf{K} := [\mathbf{k}_1, \cdots, \mathbf{k}_N], \mathbf{V} := [\mathbf{v}_1, \cdots, \mathbf{v}_N]$ where $\mathbf{q}_i, \mathbf{k}_i, \mathbf{v}_i$ are query, key, and value vectors respectively.

2. The output sequence $\mathbf{H} := [\mathbf{h}_1, \cdots, \mathbf{h}_N]$ is computed as

$$\mathbf{H}^\top = \sigma \left( \frac{\mathbf{Q}^\top \mathbf{K}}{\sqrt{D}} \right) \mathbf{V}^\top := \sigma \left( \frac{\mathbf{A}}{\sqrt{D}} \right) \mathbf{V}^\top$$

where $\sigma$ denotes the softmax function applied row-wise to $\mathbf{A} = \mathbf{Q}^\top \mathbf{K}$. Let $a_{ij}$ denote the attention scores.

Assuming linear attention, where $\sigma(\mathbf{x}^\top) = \mathbf{x}^\top$, the output for query $\mathbf{q}_i$ is:

$$\mathbf{h}_i = \sum_{j=1}^{N} \mathbf{q}_i^\top \mathbf{k}_j \cdot \mathbf{v}_j \Rightarrow \mathbf{h}_i^\top = \mathbf{W}_V \left( \sum_{j=1}^{N} \mathbf{x}_j \mathbf{x}_j^\top \right) \mathbf{W}_K^\top \mathbf{W}_Q \mathbf{x}_i. \tag{3}$$

Suppose that we cluster the token set into $C$ disjoint clusters $\{C_k\}_{k=1}^{C}$, and approximate each token $\mathbf{x}_j$ by its corresponding cluster centroid $\mathbf{x}_{k(j)}$. Define the approximated attention output by

$$\mathbf{h}_i'^\top = \mathbf{W}_V \left( \sum_{k=1}^{C} |C_k| \cdot \bar{\mathbf{x}}_k \bar{\mathbf{x}}_k^\top \right) \mathbf{W}_K^\top \mathbf{W}_Q \bar{\mathbf{x}}_{c(i)}, \tag{4}$$

where $c(k)$ and $c(i)$ denote the indices of the clusters containing tokens $\mathbf{x}_k$ and $\mathbf{x}_i$, respectively.

Now, we analyze the effect of token merging by introducing the following theorem.

**Theorem 1** (Token Merging Error Bound). *Let $\{C_k\}_{k=1}^{C}$ be a clustering of the token set $\{\mathbf{x}_j\}_{j=1}^{N}$, and let $\bar{\mathbf{x}}_k = \frac{1}{|C_k|} \sum_{j \in C_k} \mathbf{x}_j$ be the centroid of cluster $C_k$. Suppose $\mathbf{x}_i$ and all $\mathbf{x}_j \in C_k$ are replaced by their respective centroids. Then the approximation error between original and merged attention satisfies:*

$$\left\| \mathbf{h}_i - \mathbf{h}_i^{merged} \right\| \leq L_1 \cdot \left\| \mathbf{x}_i - \bar{\mathbf{x}}_{c(i)} \right\| + L_2 \cdot \sum_{k=1}^{C} \sum_{j \in C_k} \left\| \mathbf{x}_j - \bar{\mathbf{x}}_k \right\|,$$

*where $c(i)$ is the index of the cluster containing $\mathbf{x}_i$, and constants $L_1$, $L_2$ depend on network parameters and operator norms.*

*Furthermore, following Cluster Distortion Decay Assumption 1, the total error satisfies the asymptotic bound*

$$\left\| \mathbf{h}_i - \mathbf{h}_i^{merged} \right\| = \mathcal{O} \left( \frac{N}{C^{1/D_x}} \right),$$

*indicating that merging accuracy improves as the number of clusters $C$ increases.*

The proof is provided in Appendix A.

### 3.3 Empirical Structural Analysis

To validate the theoretical insight presented in Theorem 1, we begin by analyzing the structural properties of token clusters formed during the merging process. Specifically, we examine whether Fourier-based token merging results in tighter and more coherent token groupings—two key factors that directly affect the magnitude of the approximation error bound. The results shown below are conducted with a fixed truncation ratio $\tau = 0.7$.

Figure 2 presents a comparison of structural integrity between Fourier and non-Fourier merging. Each point in the figure refers to a specific merging layer. There are five merging layers for each diffusion step. The left plot reports the *cluster similarity*, defined as the average cosine similarity among token vectors within each cluster. Higher values indicate stronger internal coherence. Fourier merging

consistently achieves greater intra-cluster similarity, suggesting that it identifies more semantically aligned groups. The right plot shows the *token error within cluster*, measured as the mean $\ell_2$ distance from each token to its cluster centroid. Again, Fourier merging results in significantly lower intra-cluster variance, supporting the claim that frequency-domain groupings are geometrically compact. Looking into the individual layers in each diffusion step, for both the Fourier merging and non-Fourier merging, the cluster similarity increases as the layer becomes deeper, meaning the similarity of tokens in deeper layers is better exploited by abstracting the features of the image.

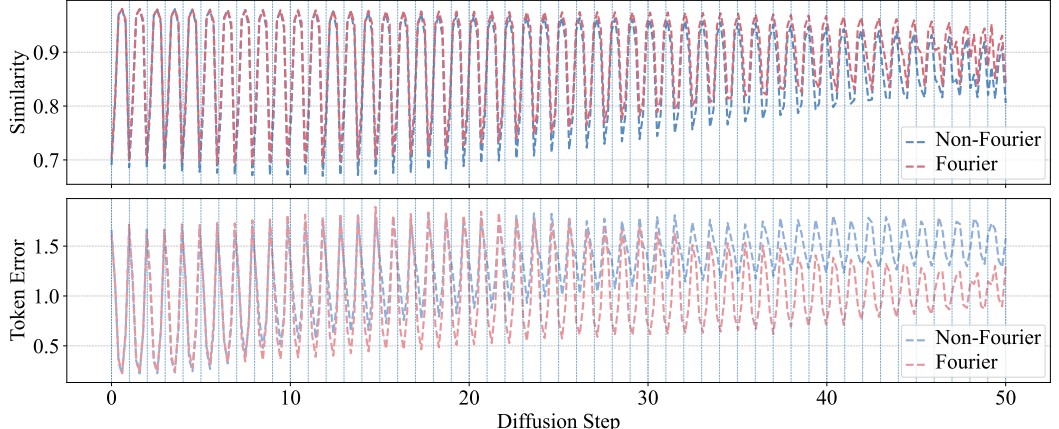

Figure 2: **Comparison of Cluster Similarity and Token Error across Diffusion Steps.** This figure presents two metrics averaged over diffusion steps to evaluate the structural behavior of token merging strategies. **Up:** The average cosine similarity among token vectors within the same cluster.Fourier-based merging maintains consistently higher similarity, indicating more coherent token grouping across the generation process. **Down:** The average token error within each cluster, defined as the mean distance between tokens and their cluster centroid. Fourier merging yields lower intra-cluster error, suggesting more compact and semantically aligned clusters. These trends highlight the advantage of frequency-based merging in preserving structural consistency over time.

To further support this observation, Table 1 reports aggregated statistics over different merging ratios for both traditional and Fourier-based methods. As the merging ratio—i.e., the degree of token count reduction per transformer block—increases, Fourier-based merging consistently maintains higher cluster similarity and lower token error. Notably, at low merging ratios (e.g., 0.1), Fourier merging achieves a 0.16 reduction in token error compared to traditional merging, while involving a similar number of active cluster centers (Here, active cluster refers to the number of unique `dst` tokens receiving merged `src` tokens at each diffusion step. For example, there are 231 unique `dst` tokens receiving merged `src` tokens). As the ratio increases, Fourier merging slightly expands the number of such active clusters, yet still demonstrates superior semantic compactness, as reflected by the lower error metric. These results suggest that Fourier-based merging distributes merges more evenly while selectively preserving semantically important structures, leading to better-structured clusters even under higher compression.

| Ratio | Traditional Token Merging | | | | Fourier Token Merging | | | |
|---|---|---|---|---|---|---|---|---|
| | Similarity ($\uparrow$) | Active Clusters ($\uparrow$) | Token Error ($\downarrow$) | Error Metric ($\downarrow$) | Similarity ($\uparrow$) | Active Clusters ($\uparrow$) | Token Error ($\downarrow$) | Error Metric ($\downarrow$) |
| 0.1 | 0.90 | **231** | 0.73 | 13.85 | **0.91** | **231** | **0.57** | **11.94** |
| 0.2 | 0.88 | 396 | **0.74** | 19.23 | **0.89** | **399** | 0.76 | **19.12** |
| 0.3 | 0.87 | 507 | 0.98 | 33.61 | **0.89** | **523** | **0.86** | **27.11** |
| 0.4 | 0.87 | **624** | 1.23 | 46.96 | **0.88** | 605 | **0.98** | **40.56** |
| 0.5 | **0.86** | 706 | **1.32** | 59.29 | **0.86** | **716** | 1.38 | **58.82** |

Table 1: Comparison between non-Fourier and Fourier token merging across different merge ratios. Bold numbers indicate better performance ($\uparrow$: higher is better; $\downarrow$: lower is better).

In Figure 3, we further investigate cluster statistics across diffusion steps. The left subplot reports the number of *active cluster centers* (i.e., `dst` tokens) that the `src` tokens are merged into at each step.

This effectively reflects the distribution spread of token merges and indicates the level of compression granularity. The shaded region denotes the standard deviation across five runs. Interestingly, despite its structure-preserving nature, Fourier-based merging results in a comparable or even higher number of utilized cluster centers than its non-Fourier counterpart, suggesting broader merge distribution and reduced concentration bias.

The right subplot depicts an aggregate *error metric within cluster*, designed to reflect intra-cluster dispersion as it appears in the second term of the inequality in Theorem 1. Although traditional merging performs clustering directly in the time domain, Fourier-based merging surprisingly achieves lower within-cluster error throughout the generation trajectory.

This apparent contradiction arises from the fact that Fourier clustering operates on a denoised, low-frequency representation of the tokens, suppressing high-frequency fluctuations that often introduce spurious local variation. As a result, the clusters it produces are more semantically coherent and robust to transient noise. Thus, even though Fourier merging makes clustering decisions in a compressed space, the resulting assignments tend to yield lower dispersion in the original domain.

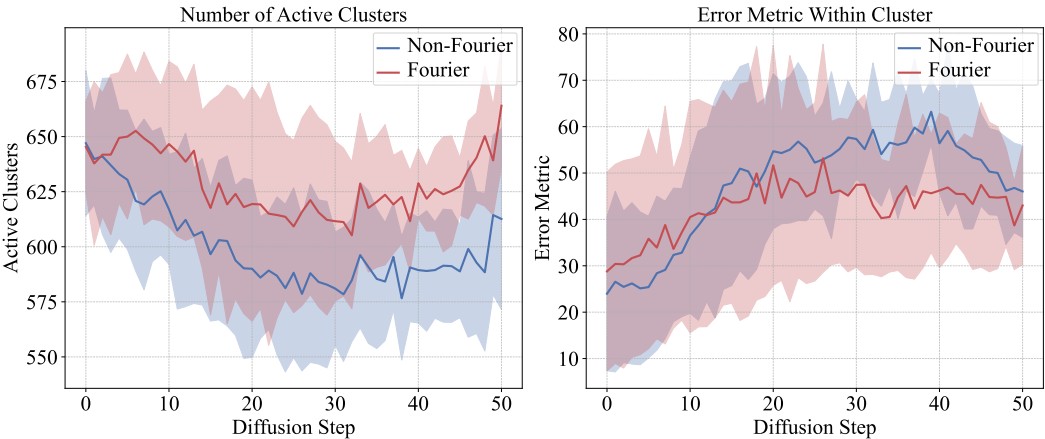

Figure 3: **Evolution of Error Metrics and Number of Active Cluster Centers across Diffusion Steps.** This figure analyzes the dynamics of token merging through two global metrics. **Left:** Number of unique `dst` tokens receiving merged `src` tokens at each diffusion step. This reflects the effective number of cluster centers actively used during token merging. Fourier-based merging exhibits a more stable and often broader distribution of merges, indicating smoother and semantically consistent clustering behavior. **Right:** A global error metric measuring total intra-cluster variation. The Fourier merging consistently achieves lower error across the generation timeline, indicating better information retention. Together, these plots validate the efficiency and accuracy benefits of incorporating Fourier-domain priors in token merging strategies.

Together, these results confirm that frequency-domain merging yields clusters with higher semantic coherence and reduced variance—two properties that, according to our theoretical analysis, directly reduce the approximation error in attention computations.

## 4 Adaptive Fourier Token Merging Scheme

### 4.1 Cluster Visualization Analysis

To better understand the behavior of frequency-based token merging, we begin with a qualitative visualization of clustering results. Figure 4 presents examples from two cat images in the CIFAR-10 dataset, where clustering is applied to the red channel of the original $32 \times 32$ images. For each pixel, we extract a $5 \times 5$ local patch centered at that pixel and treat it as a token. The figure shows the largest 10 clusters obtained using both Fourier and non-Fourier token merging, visualized as 2D scatter plots.

We observe that Fourier-based clustering produces more structured and semantically coherent clusters. By suppressing high-frequency components, it is able to group together patches that are visually similar in texture or pattern, even when their raw pixel values differ significantly due to noise or local variance. This indicates that frequency-domain clustering can effectively abstract away irrelevant variations, yielding more meaningful token groupings.

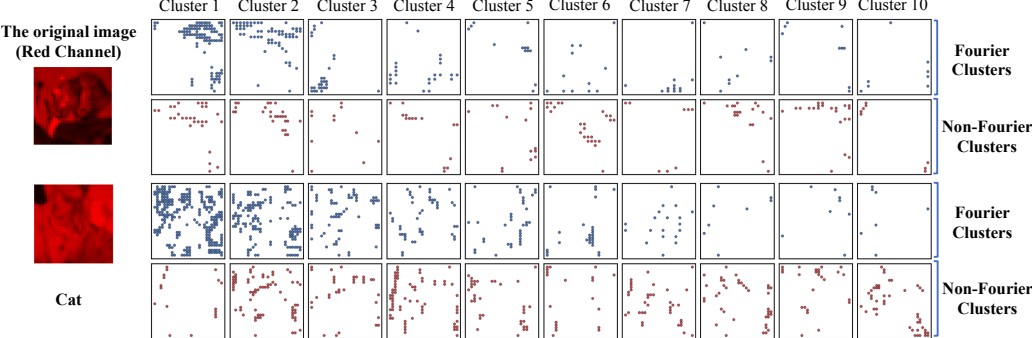

Figure 4: Visualization of the top 10 clusters for Fourier and non-Fourier merging on two CIFAR-10 cat images (red channel only). Each point corresponds to a $5 \times 5$ local patch treated as a token belonging to a particular cluster. Fourier-based clustering yields denser and more structured groupings.

To assess whether these benefits generalize to deeper representations, we extend our analysis in Appendix B (Figure 9) to intermediate activations from a CifarNet-like model (e.g., outputs from the second convolutional layer). Unlike raw pixel inputs, these activations resemble multi-channel heatmaps with more diffuse spatial organization.

In this higher-level feature space, we find that clustering results remain structured, but now prioritize semantic similarity over local texture. Clusters appear less periodic and more abstract, indicating a shift in the type of information being preserved.

These findings suggest that the optimal degree of frequency truncation should be input-dependent: For low-level features (e.g., raw RGB patches), retaining more frequency information helps capture spatial regularities. For high-level activations, more aggressive truncation can be beneficial, suppressing noise and redundancy to highlight semantically relevant patterns.

### 4.2 Adaptive Scheme for Truncation Ratio Scheduling

Motivated by the above analysis, we introduce an adaptive truncation scheme that dynamically adjusts the truncation ratio based on diffusion timestep and layer depth—two factors that affect token complexity and noise level.

First, we adapt truncation along the diffusion step axis. In early diffusion steps, the input representations are highly stochastic and noisy. This favors stronger truncation to eliminate high-frequency noise and improve clustering robustness. As generation proceeds and signals become more refined, we gradually reduce the truncation ratio to preserve finer details.

Second, we adapt along the layer depth within the network. Early layers tend to encode low-level details such as texture and edges, which are sensitive to frequency suppression. To retain these features, we apply a higher truncation ratio (i.e., less truncation). In deeper layers, the tokens become more abstract and semantically structured, allowing for more aggressive frequency pruning without degrading cluster coherence.

Formally, we model the truncation ratio $\tau$ as a function of timestep $t$ and layer index $\ell$, normalized within their respective ranges:

$$\tau(t, \ell) = \text{interpolate}\left(\tau_{\min}, \tau_{\max}; \lambda_t \cdot \text{norm}(t) + \lambda_\ell \cdot \text{norm}(\ell)\right)$$

where $\lambda_t$ and $\lambda_\ell$ control the relative importance of timestep and layer depth, and $\tau_{\min}, \tau_{\max}$ define the bounds of the truncation ratio. These are considered as hyperparameters.

This adaptive scheduling mechanism ensures that *Fourier Token Merging* remains sensitive to the structural demands of each layer and each stage of the generation process, leading to improved quality–latency tradeoffs across the board.

## 5 Experiments

We conduct a series of experiments on the *Fourier Token Merging* to validate its efficacy of for efficient image generation.

## 5.1 Experimental Setup

**Methodology.** To evaluate the effectiveness of *Fourier Token Merging*, we integrate the proposed frequency-based method into Stable Diffusion (SD) (CC-BY 4.0) models [21]. We experiment with various merging ratios—i.e., different degrees of token count reduction per transformer block—to assess performance under different merging configurations. Results are compared against those of the original *Token Merging* baseline. The *Fourier Token Merging* is applied to transformer blocks with a downsampling level of at most 1, covering a total of five layers. We select $\tau_{\min}$ in $\{0.3, 0.4, 0.5\}$ and $\tau_{\max}$ in $\{0.6, 0.8, 1.0\}$, respectively, and traverse $\lambda_t = 1 - \lambda_\ell$ in $\{0, 0.1, \cdots, 1.0\}$ and report the best result.

**Platform.** Experiments are conducted on a system equipped with a 12-core (8 Performance-cores and 4 Efficient-cores) Intel Core i7-12700K CPU operating at a base frequency of 3.60GHz, 128GB of RAM, and an NVIDIA GeForce RTX 4090 GPU with 24GB of GDDR6X memory. The GPU features 16,384 CUDA cores based on the Ada Lovelace architecture, offering a memory bandwidth of up to 1,008GB/s.

**Workloads.** The goal of *Fourier Token Merging* is to achieve better performance in SD models while maintaining generation quality close to that of the full-token baseline. To evaluate this, we sample two images per class from all 1,000 ImageNet (CC-BY 4.0) categories, using prompts of the form *"a photo of a CLASS"*. Comparisons are made with a fixed random seed to ensure consistency across runs.

We use the Fréchet Inception Distance (FID) [9] to assess the distributional similarity between images generated by the diffusion model and 5,000 class-balanced ImageNet-1k val examples. In addition, LPIPS [32] is used to measure pairwise perceptual similarity between images generated by the full model and the ones that use token merging. while MS-SSIM [27, 28] quantifies pairwise structural consistency. Both LPIPS and MS-SSIM are computed at the image level to capture deviations from the full-token baseline under a fixed random seed.

## 5.2 Experimental Results

We evaluate *Fourier Token Merging* by measuring its impact on both image quality and generation efficiency. Figure 5 reports latency against three metrics: FID, LPIPS, and MS-SSIM.

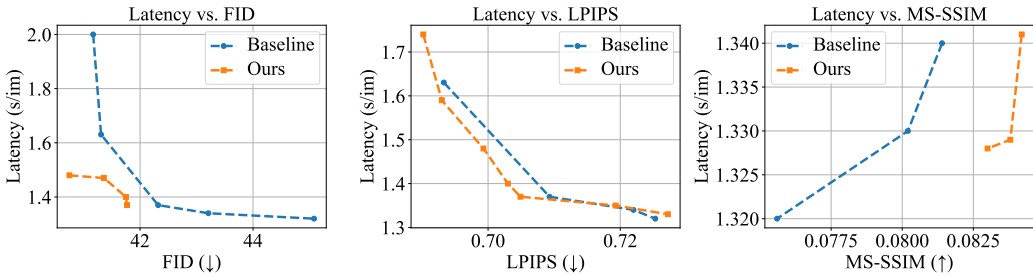

Figure 5: Comparison of latency versus image quality metrics between the baseline and our method. The different points on a curve correspond to different token merging ratios. **Left**: FID vs. latency, showing that our method achieves lower latency for comparable or better FID. **Middle**: LPIPS vs. latency, demonstrating improved perceptual similarity over baseline. **Right**: MS-SSIM vs. latency, indicating stronger structural consistency with minimal latency overhead. Note that Pareto frontiers are selected from a set of hyperparameter settings, and therefore differ across subplot. Overall, our approach consistently improves the quality-efficiency tradeoff across multiple evaluation metrics.

**FID.** Our method achieves comparable or improved FID relative to the baseline across all configurations, while consistently reducing generation latency. This indicates that frequency-domain merging preserves the global semantics of generated images.

**LPIPS.** Despite aggressive compression, *Fourier Token Merging* maintains perceptual similarity on par with the baseline. In several cases, it achieves lower LPIPS at lower latency, suggesting improved retention of local visual features.

**MS-SSIM.** We observe consistently higher MS-SSIM scores with minimal latency overhead, indicating stronger structural coherence in the generated outputs. This aligns with the hypothesis that frequency-based grouping enhances token alignment.

Across all metrics, *Fourier Token Merging* delivers favorable quality-efficiency tradeoffs, validating the theoretical benefits of frequency-aware token reduction in diffusion models.

## 5.3 Ablation Study

To better understand the contribution of each component in our design, we perform an ablation study across two axes: (i) the effect of the truncation ratio under fixed token merging ratios, and (ii) the impact of using Fourier-based clustering versus alternative frequency-domain, such as wavelet transforms.

**Effect of Truncation Ratio.** To investigate the influence of the truncation ratio $\tau$, we conduct a controlled ablation by varying the fixed $\tau \in 0.2, 0.3, \ldots, 1.0$ throughout the model across five fixed merging ratios (0.1 to 0.5). The resulting FID scores and latencies are reported in Figure 6. While the latency remains roughly the same across the truncation ratio, the FID score can change with different levels of truncation. We note that the optimal truncation level depends on the merging aggressiveness. When the merging ratio is low (i.e., fewer tokens are merged), preserving high-frequency information helps maintain generation fidelity. However, as merging becomes more aggressive, discarding more high-frequency components improves output quality. This suggests that low-frequency clustering becomes increasingly beneficial under stronger compression, possibly due to its robustness to local noise and redundancy.

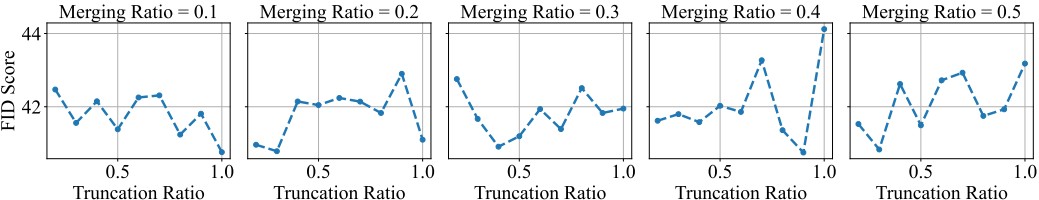

Figure 6: FID vs. Truncation Ratio under varying token merging ratios.

**Wavelet vs. Fourier**. We further compare the proposed DFT-based approach against a wavelet-based clustering scheme (Haar wavelet). Results in Figure 7 indicate that wavelet-based merging exhibits higher latency and inferior FID compared to our method. This gap is mainly attributed to the extra overhead from multi-resolution decomposition and the lack of efficient GPU methods to implement wavelet transformation. Our Fourier approach, in contrast, allows for efficient transformation and preserves salient global structure, leading to more favorable accuracy-efficiency tradeoffs.

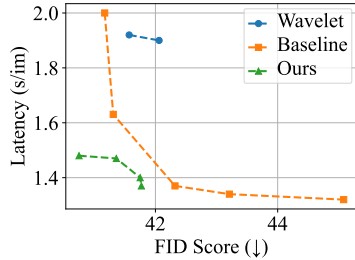

Figure 7: Wavelet vs. Fourier Token Merging: FID vs. Latency.

**Real-coefficients of Fourier tokens vs. full complex tokens.** For efficiency, we adopt only the real coefficients of the Fourier tokens. Empirically, using the real part consistently achieves comparable or superior performance. For example, it attains 84.33% accuracy, slightly outperforming both the absolute-value representation (84.29%) and the full complex representation (84%).

## 5.4 Results for Generalizability

**Generalization to Diffusion Transformers (DiTs).** Our Fourier Token Merging method aims for broader applicability. Existing token merging methods often struggle with DiT's specialized architecture and ROPE embeddings. Specifically, our implementation of Fourier Token Merging for the DiT architecture, which handles the complex structure of Joint/Single transformers, follows the integration

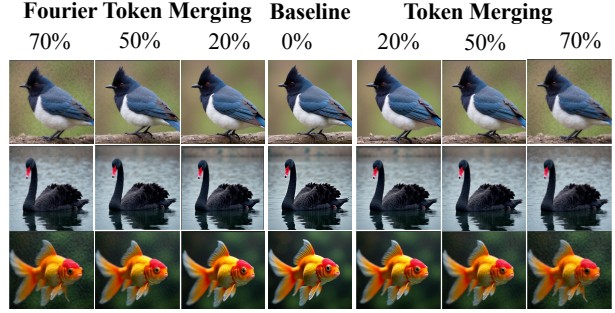

Figure 8: Visualization of Token Merging and Fourier Token Merging on the FLUX model.

strategy introduced by Token Merging with Attention (ToMA) [14]. We have done experiments on DiT models, specifically FLux.1-dev, tailoring FTM to their specific transformer blocks. Results show benefits. Even with a large merging ratio (0.75), we maintain high image quality (1.83 FID increase) while achieving over 20% latency reduction. This suggests FTM's strong generalization capabilities to diverse diffusion models, including DiT-like architectures, distinguishing our approach and highlighting its potential for current foundation models.

**Generalization to Image Classification.** We experimented our Fourier Token Merging in ViT models. Results show that our method generalizes well on image classification tasks. For ImageNet val-1k dataset on the 4090 machine, with the same level of throughput (e.g. around 2000 im/s), our method increases the accuracy by 0.7 point (from 83.09% to 83.79%). With the same level of accuracy, our method achieves $1.2 \times$ speedups.

Regarding deployment efficiency, our empirical measurements confirm that the overhead is minimal. The wall-clock time for the FFT is under 0.1 ms on average, which accounts for only 0.25% to 0.4% of the total inference time.

Overall, this study highlights the value of frequency-domain structural cues and validates our design choices in achieving compression with minimal quality degradation.

# 6 Conclusion

This work introduces *Fourier Token Merging*, a novel method for accelerating image generation by exploiting token redundancy in the frequency domain. By transforming tokens via the Discrete Fourier Transform and performing clustering in the low-frequency subspace, our method improves the semantic alignment and compactness of token groups, leading to lower approximation error in self-attention computations. Theoretical analysis formalizes the error bounds introduced by token merging, and extensive empirical studies across structural metrics, attention distortion, and image quality evaluations (FID, LPIPS, MS-SSIM) confirm the effectiveness of our approach.

Compared to baseline token merging methods, *Fourier Token Merging* achieves up to 25% latency reduction while maintaining or even improving generation quality. Ablation studies further demonstrate the importance of frequency-aware clustering and validate the impact of truncation ratios under varying merging aggressiveness. We also explore adaptive schemes that adjust frequency truncation based on layer depth and diffusion timestep, offering a flexible and robust extension to the proposed framework. One limitation is that Fourier transform may be computationally intensive for resource-constrained devices. Its societal impacts are neutral.

Overall, our results highlight the potential of leveraging frequency-domain structure for efficient generative modeling, and provide a modular, plug-and-play solution applicable to a wide range of diffusion-based image generation systems.

# 7 Acknowledgement

We thank the anonymous reviewers for the constructive comments. We would also like to thank Wenbo Lu for sharing their implementation code for Token Merging with Attention (ToMA), which was instrumental in enabling our Flux.1-dev experiments. This material is based upon work supported by the National Institutes of Health (NIH) under Grant No. 1R01HD108473-01 and National Science Foundation (NSF) under Grant No. OAC-2417850 and P24-001771. Any opinions, findings, and conclusions or recommendations expressed in this material are those of the authors and do not necessarily reflect the views of NIH or NSF.

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

# A Proof of Theorem 1

**Assumption 1.** *Although the clustering method used in our token merging strategy is not necessarily k-means, we assume that it produces clusters with distortion comparable to those obtained by k-means. That is, we assume the mean-squared intra-cluster error satisfies the standard quantization rate:*

$$MSE_{cluster} := \frac{1}{N} \sum_{j=1}^{N} \|\mathbf{x}_j - \bar{\mathbf{x}}_{c(j)}\|^2 = \mathcal{O}\left(\frac{1}{C^{2/d_x}}\right),$$

*where $d_x$ is the input feature dimension, and $C$ is the number of clusters. This assumption is commonly satisfied under mild regularity conditions when the data lies on a compact manifold or is sampled from a distribution with bounded support.*

*Proof.* We express the attention output as:

$$\mathbf{h}_i = \sum_{j=1}^{N} f(\mathbf{x}_i, \mathbf{x}_j), \quad \text{where} \quad f(\mathbf{x}_i, \mathbf{x}_j) := (\mathbf{W}_Q \mathbf{x}_i)^\top (\mathbf{W}_K \mathbf{x}_j) \cdot \mathbf{W}_V \mathbf{x}_j.$$

Under token merging, all $\mathbf{x}_j \in C_k$ are replaced by $\bar{\mathbf{x}}_k$, and the query $\mathbf{x}_i \in C_{c(i)}$ is replaced by $\bar{\mathbf{x}}_{c(i)}$. The merged attention output is:

$$\mathbf{h}_i^{merged} = \sum_{k=1}^{K} \sum_{j \in C_k} f(\bar{\mathbf{x}}_{c(i)}, \bar{\mathbf{x}}_k)$$

The total error is:

$$\delta \mathbf{h}_i := \mathbf{h}_i^{merged} - \mathbf{h}_i = \sum_{k=1}^{K} \sum_{j \in C_k} \left( f(\bar{\mathbf{x}}_{c(i)}, \bar{\mathbf{x}}_k) - f(\mathbf{x}_i, \mathbf{x}_j) \right)$$

For each pair $(i, j)$, we apply first-order Taylor expansion around $(\mathbf{x}_i, \mathbf{x}_j)$:

$$f(\bar{\mathbf{x}}_{c(i)}, \bar{\mathbf{x}}_k) - f(\mathbf{x}_i, \mathbf{x}_j) \approx \nabla_{\mathbf{x}_i} f \cdot (\bar{\mathbf{x}}_{c(i)} - \mathbf{x}_i) + \nabla_{\mathbf{x}_j} f \cdot (\bar{\mathbf{x}}_k - \mathbf{x}_j)$$

Taking the norm and summing over all $j$, we obtain:

$$\|\delta \mathbf{h}_i\| \leq \sum_{k=1}^{K} \sum_{j \in C_k} \left( \|\nabla_{\mathbf{x}_i} f\| \cdot \|\bar{\mathbf{x}}_{c(i)} - \mathbf{x}_i\| + \|\nabla_{\mathbf{x}_j} f\| \cdot \|\mathbf{x}_j - \bar{\mathbf{x}}_k\| \right)$$

Bounding gradients by constants $L_1, L_2$, we arrive at:

$$\|\delta \mathbf{h}_i\| \leq L_1 \cdot \|\bar{\mathbf{x}}_{c(i)} - \mathbf{x}_i\| + L_2 \cdot \sum_{k=1}^{K} \sum_{j \in C_k} \|\mathbf{x}_j - \bar{\mathbf{x}}_k\|$$

We then look into the second half of the theorem. Following 3 and 4, the total approximation error can be decomposed as

$$\Delta \mathbf{h}_i^\top = \mathbf{h}_i^\top - \mathbf{h}_i'^\top = \mathbf{W}_V \left[ (\Sigma_x - \Sigma_x') \mathbf{W}_K^\top \mathbf{W}_Q \mathbf{x}_i + \Sigma_x' \mathbf{W}_K^\top \mathbf{W}_Q (\mathbf{x}_i - \bar{\mathbf{x}}_{c(i)}) \right],$$

where $\Sigma_x := \sum_{j=1}^{N} \mathbf{x}_j \mathbf{x}_j^\top$ and $\Sigma_x' := \sum_{k=1}^{C} |C_k| \bar{\mathbf{x}}_k \bar{\mathbf{x}}_k^\top$.

We now bound each term individually. Let $\mathbf{e}_j := \mathbf{x}_j - \bar{\mathbf{x}}_{c(j)}$ denote the clustering error. Then

$$\|\Sigma_x - \Sigma_x'\|_F \leq 2RN \sqrt{\frac{1}{N} \sum_{j=1}^{N} \|\mathbf{e}_j\|^2} = 2RN \cdot \sqrt{MSE_{cluster}}.$$

If the input distribution satisfies mild regularity conditions (e.g., bounded support or sub-Gaussian), classical results from quantization theory imply

$$\text{MSE}_{\text{cluster}} = \mathcal{O}\left(\frac{1}{C^{2/D_x}}\right) \Rightarrow \|\Sigma_x - \Sigma'_x\|_F = \mathcal{O}\left(\frac{N}{C^{1/D_x}}\right).$$

Meanwhile, the second term satisfies

$$\|\mathbf{x}_i - \bar{\mathbf{x}}_{c(i)}\| \le \max_j \|\mathbf{e}_j\| \le \sqrt{\text{MSE}_{\text{cluster}}} = \mathcal{O}\left(\frac{1}{C^{1/D_x}}\right).$$

Combining both, we have

$$\|\mathbf{h}_i^\top - \mathbf{h}_i'^\top\| \le \|\mathbf{W}_V\| \cdot \left(\mathcal{O}\left(\frac{N}{C^{1/D_x}}\right) \cdot \|\mathbf{W}_K^\top \mathbf{W}_Q \mathbf{x}_i\| + \mathcal{O}(1) \cdot \|\mathbf{W}_K^\top \mathbf{W}_Q\| \cdot \frac{1}{C^{1/D_x}}\right),$$

which simplifies to

$$\|\mathbf{h}_i - \mathbf{h}_i'\| = \mathcal{O}\left(\frac{N}{C^{1/D_x}}\right).$$

This concludes the proof. $\qquad\square$

# B  Additional Clustering Visualization

We show the clustering visualization when the inputs are the intermediate representation.

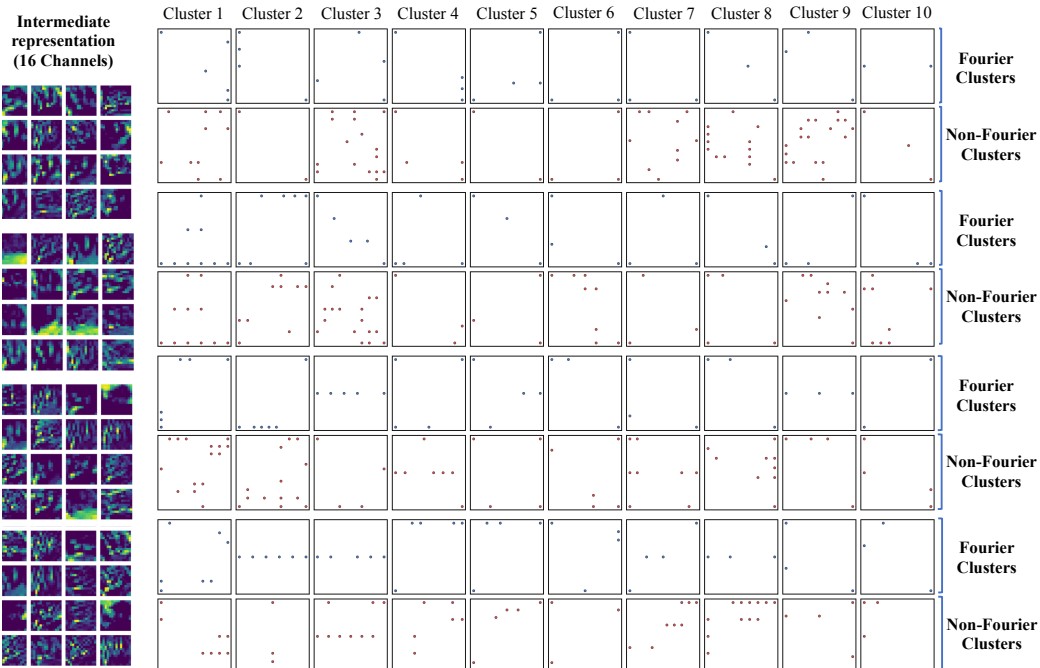

Figure 9: Visualization of the top 10 clusters for Fourier and non-Fourier merging on intermediate representations. Same setting as in Figure 4. Fourier-based clustering yields denser and more structured groupings.

# C  Additional Related Work

Numerous methods have been proposed to reduce the computational cost of transformer-based models by minimizing token-level redundancy [1, 4, 7, 10, 14, 15, 17, 29, 30]. Among the most prominent are

Token Merging (ToMe) [2], DynamicViT [18], and AdaViT [16], which dynamically prune or merge tokens based on content similarity or learned importance. These approaches significantly improve inference efficiency. However, they operate exclusively in the time domain and thus do not exploit the latent redundancy that may be more readily revealed through frequency-domain transformations.

**Token Reduction in Vision Transformers.** Token-level acceleration strategies have received substantial attention in recent years. ToMe [2] reduces the number of tokens by clustering and merging similar ones during inference. Beyond pruning, other notable works focus on reorganizing token representations for efficiency. For instance, TC-Former [31] introduces a human-centric clustering transformer that merges tokens based on content importance, while EViT [13] employs progressive token pruning or early exiting by leveraging attention rollout and token significance. Touvron et al. [24] introduce a distillation-based approach for Vision Transformers, where a specialized token is used to distill and condense teacher-model representations into a task-specific embedding, while Patch Slimming [23] reduces spatial redundancy through aggressive downsampling.

**Frequency-domain Representations.** Compared to the wealth of time-domain methods, only a few studies have explored frequency-domain approaches for model acceleration. FNet [12] replaces attention mechanisms with Fourier transforms in NLP models, offering a lightweight alternative to self-attention. Spectral pooling [20] leverages low-pass filtering in the frequency domain to compress CNN activations. GFNet [19] highlights the untapped potential of frequency representations by replacing self-attention with learnable global filters in the frequency domain.

**Additional Comparison to SOTA.** We included the results of several SOTA baselines as follows. AT-EDM [25] is an SOTA token pruning method that offers $1.13\times$ speedup over ToMe but loses quality. Our method, at a similar $1.1\times$ speedup, actually decreases FID (40.75 vs. 41.31), showing better quality-speed trade-off. Other token Downsampling method [22] helps only at very high merging ratios (e.g., 0.89), leading to longer latency at lower ratios (e.g., 0.75). In contrast, our method consistently achieves speedups across various ratios, e.g., $1.3\times$ at 41.8 FID.

