# OpenReview forum: "Fourier Token Merging: Understanding and Capitalizing Frequency Domain for Efficient Image Generation"
_NeurIPS.cc/2025/Conference — NeurIPS 2025 poster_

### Official Review · Reviewer_xhJ6 · 2025-06-20

**Clarity:** 2
**Significance:** 3
**Originality:** 3
**Rating:** 4
**Confidence:** 3

**Summary:**

The paper presents Fourier Token Merging, which improves image generation efficiency by using frequency domain clustering to reduce token redundancy. Unlike traditional methods in the time domain, this approach leverages Fourier transforms to better exploit redundancy, reducing latency by up to 25% in experiments with the Stable Diffusion model, while maintaining image quality. This contributes a novel technique for optimizing neural network computations in image generation.

**Questions:**

1. I think the amount of experiments in this work is not enough, especially more diffussion methods should be used for experiments instead of just one SD model.
2. Figure 1 is not clear, and it is difficult to intuitively see the execution process of the Fourier Merging Module. It would be better to have a figure to explain the merging process in detail.
3. In line 84, "only the lowest τd frequency", what is τd? It seems that there is a typo here.
4. What is the connection between 3.2 and 3.1? A clear expression may be needed here.
5. The writing of the Modular Integration section seemed confusing, and although there were descriptions of the advantages of the approach, I thought they were just common sight and did not help me understand how the proposed approach was effective.

**Ethical Concerns:**

["NO or VERY MINOR ethics concerns only"]

**Final Justification:**

Thank you for authors' reply. I keep my original opinion.

**Limitations:**

1. Although this work contains a lot of quantitative results, it lacks image visualization comparison.

**Paper Formatting Concerns:**

No formatting concern is found.

**Quality:**

3

**Strengths And Weaknesses:**

1. By transforming tokens into the frequency domain, the method better leverages inherent redundancy after de-correlation, leading to more efficient token merging.
2. Frequency domain clustering allows for more precise identification and merging of similar tokens, compared to traditional time domain methods.
3. Experiments show up to a 25% reduction in latency during image generation processes.

---

> ### Author Rebuttal · Authors · 2025-07-31
>
> Dear Reviewer,
>
> Thank you for the insightful comments. We address the following concerns:
>
> > **Comment 1: I think the amount of experiments in this work is not enough, especially more diffussion methods should be used for experiments instead of just one SD model.**
>
> Response: We applied our method to more models as follows.
> We experimented our Fourier Token Merging in ViT models.
> Results show that our method generalizes well on image classification tasks.
> For ImageNet val-1k dataset on the 4090 machine, with the same level of throughput (e.g. around 2000 im/s), our method increases the accuracy by 0.7 point (from 83.0\% to 83.79\%).
> With the same level of accuracy, our method achieves 1.2$\times$ speedups.
>
> Aside from the image classfication experiment, we also include results for Diffusion Transformer (DiT) models.
> Specifically, we have done experiments on DiT models, tailoring FTM to their specific transformer blocks. Results are promising: even with a large merging ratio (0.75), we maintain high image quality (1.83 FID increase) while achieving over 20\% latency reduction.
> This suggests FTM's strong generalization capabilities to diverse diffusion models, including DiT-like architectures, distinguishing our approach and highlighting its potential for current foundation models.
>
> We will add those results into the final version.
>
> > **Comment 2: Figure 1 is not clear, and it is difficult to intuitively see the execution process of the Fourier Merging Module. It would be better to have a figure to explain the merging process in detail.**
>
> Response: We will make clear the merging process through image illustration as detailed in Section 3.1.
>
> > **Comment 3: In line 84, ``only the lowest $\tau d$ frequency", what is $\tau d$? It seems that there is a typo here.**
>
> Response: As indicated earlier in the paragraph, $d$ is the dimension of the token **x** and $\tau$ is the truncation ratio, so $\tau d$ is $\tau \times d$ which refers to the dimension we keep after truncating on the frequency domain. We will add the clarification into the text to avoid any confusions.
>
> > **Comment 4: What is the connection between 3.2 and 3.1? A clear expression may be needed here.**
>
> Response: Token Merging itself is an approximation computing method and brings in error for the image generation process.
> Section 3.1 discusses our main idea of introducing the Fourier Merging Module into the Token Merging pipeline.
> A theoretical interpretation and intuition behind the Fourier Token Merging is then explained in Section 3.2 and show the potential of achieving smaller approximation error compared to the original Token Merging.
>
> > **Comment 5: The writing of the Modular Integration section seemed confusing, and although there were descriptions of the advantages of the approach, I thought they were just common sight and did not help me understand how the proposed approach was effective.**
>
> Response: Our idea is to cluster (merging) tokens in the frequency domain so as to maximize its decorrelation ability.
> We show both theoretically that our method gives smaller approximation errors compared to the original method (Section 3.2) and empirically that our method merges tokens that have higher similarity scores (Section 3.3).
> Section 4 shows that Fourier-based clustering produces more structured and semantically coherent clusters.
> We further evaluate end-to-end image generation and find our method not only increases the image generation quality but also achieves speedup compared to the original Token Merging method.

---

### Official Review · Reviewer_qJuw · 2025-06-24

**Clarity:** 3
**Significance:** 2
**Originality:** 3
**Rating:** 4
**Confidence:** 4

**Summary:**

The paper studies fourier token merging to accelerate image generation in diffusion models. It leverages frequency domain for cluster and token reduction. The proposed method results up to 25% reduction in generation latency with no loss in the performance. The author also provides analysis showing the insights of the proposed method.

**Questions:**

See Weaknesses above.

**Ethical Concerns:**

["NO or VERY MINOR ethics concerns only"]

**Final Justification:**

Most of my concerns have been addressed in the rebuttal.

**Limitations:**

Yes.

**Paper Formatting Concerns:**

No.

**Quality:**

3

**Strengths And Weaknesses:**

**Strengths**:
- The use of frequency domain for token merging is interesting.
- The motivation is clear for the efficiency of generation process.
- The proposed method is shown to be effective.

**Weaknesses**:
- The current method is evaluated only on diffusion models with U-Net architecture. Applicability to up-to-date architectures like Diffusion Transformer is not demonstrated.
- The method introduces more hyper-parameters and the token merging process is highly affected by this hyper-parameters which may require more tuning during the training process.
- The improvement is not significant. In Figure 5, The FID and latency improvement is not significant when compared to the baseline.

---

> ### Author Rebuttal · Authors · 2025-07-31
>
> Dear Reviewer,
>
> Thank you for the insightful comments. We address the following concerns:
>
> > **Comment 1: The current method is evaluated only on diffusion models with U-Net architecture. Applicability to up-to-date architectures like Diffusion Transformer is not demonstrated.**
>
> Response: While we primarily evaluated on Stable Diffusion, our Fourier Token Merging method w for e applicability. Existing token merging methods often struggle with DiT's specialized architecture and RoPE embeddings.
> We have done experiments on DiT models, tailoring FTM to their specific transformer blocks. Results are promising: even with a large merging ratio (0.75), we maintain high image quality (1.83 FID increase) while achieving over 20\% latency reduction.
> This suggests FTM's strong generalization capabilities to diverse diffusion models, including DiT-like architectures, distinguishing our approach and highlighting its potential for current foundation models. We will add the results into the final version.
>
> > **Comment 2: The method introduces more hyper-parameters and the token merging process is highly affected by this hyper-parameters which may require more tuning during the training process.**
>
> Response: Our Fourier Token Merging method uses more hyperparameters, but this provides crucial control over the efficiency-quality trade-off, letting users customize for their needs (e.g., merging ratio).
> Optimal values for parameters like truncation ratios will be included in the config file.
> Crucially, as Fourier Token Merging is an off-the-shelf, training-free method, parameter tuning is highly efficient, allowing quick experimentation and rapid optimization for applications.
>
> > **Comment 3: The improvement is not significant. In Figure 5, The FID and latency improvement is not significant when compared to the baseline.**
>
> Response: Our method not only reduces latency but also increases the image quality.
> Figure 5 clearly illustrates that for the same level of image quality (e.g., around 41.5 FID), our FTM method achieves approximately 1.5$\times$ speedup compared to ToMeSD. Conversely, if we target the same inference speed (e.g., around 1.38 seconds per image), our method yields a 0.5 lower FID score, indicating superior image quality.

---

### Official Review · Reviewer_kCHM · 2025-07-01

**Clarity:** 3
**Significance:** 3
**Originality:** 4
**Rating:** 4
**Confidence:** 3

**Summary:**

This paper proposes Fourier Token Merging to speed up image generation. It transforms tokens into the frequency domain and clusters them using low-frequency information. This makes the tokens more compact and semantically aligned. The Fourier Token Merging reduces errors in self-attention and experiments on Stable Diffusion show up to 25% lower latency without losing image quality. The theoretical analysis is sufficient and the experiments show that the method is lightweight and easy to use with existing models.

**Questions:**

Will the code for this paper be open-sourced? It is recommended to do so, as it would greatly benefit and advance the community.

**Ethical Concerns:**

["NO or VERY MINOR ethics concerns only"]

**Final Justification:**

The author‘s rebuttal has addressed my concerns. The additional experiments to validate the generalizability of the proposed Fourier Token Merging (FTM) module are well-executed and convincing. I will keep my original rate.

**Limitations:**

Please refer to the weakness part for rebuttal.

**Paper Formatting Concerns:**

The paper dose not contain paper formatting problem.

**Quality:**

3

**Strengths And Weaknesses:**

Strengths:

(1) The idea of performing token merging in the frequency domain is well-motivated. It extends beyond traditional spatial clustering by exploiting the decorrelation properties of DFT.

(2) The formal error bound analysis for token merging is good and the reasons about why frequency-domain clustering reduces approximation error make sense.

(3) In the experiment part, the method shows consistent improvements across FID, LPIPS and MS-SSIM metrics with significant inference speedups.

Weaknesses:

(1) Figure 1 needs to be refined. The clarity and aesthetics of this figure could be improved.

(2) The paper does not provide any visual comparison of generated images between the proposed Fourier Token Merging and the baseline method. Since the primary goal is to preserve image quality while reducing computational cost, qualitative results are crucial to verify that the visual fidelity and semantics are indeed retained. Including such examples would strengthen the empirical claims.

(3) The Fourier Token Merging indicates a plug-and-play solution, but the experiment is insufficient. It only conducts experiments based on stable diffusion model. In my opinion,  if a method clarifies that the method is a plug-and-play module, it needs to verify the proposed module into different backbones for a wider evaluation of the effectiveness.

---

> ### Author Rebuttal · Authors · 2025-07-31
>
> Dear Reviewer,
>
> Thank you for the insightful comments. We address the following concerns:
>
> > **Comment 1: Figure 1 needs to be refined. The clarity and aesthetics of this figure could be improved.**
>
> Response: We will enhance the figure.
>
> > **Comment 2: The paper does not provide any visual comparison of generated images between the proposed Fourier Token Merging and the baseline method. Since the primary goal is to preserve image quality while reducing computational cost, qualitative results are crucial to verify that the visual fidelity and semantics are indeed retained. Including such examples would strengthen the empirical claims.**
>
> Response: We'll address this by adding a dedicated section with visual comparisons of images generated using our method at various merging ratios, along with those from the original Stable Diffusion and the ToMeSD adapted model. It will clearly illustrate the quality retention of our method, even with higher merging ratios.
>
> > **Comment 3: he Fourier Token Merging indicates a plug-and-play solution, but the experiment is insufficient. It only conducts experiments based on stable diffusion model. In my opinion, if a method clarifies that the method is a plug-and-play module, it needs to verify the proposed module into different backbones for a wider evaluation of the effectiveness.**
>
> Response: We applied our method to more models as follows.
> We experimented our Fourier Token Merging in ViT models.
> Results show that our method generalizes well on image classification tasks.
> For ImageNet val-1k dataset on the 4090 machine, with the same level of throughput (e.g. around 2000 im/s), our method increases the accuracy by 0.7 point (from 83.0\% to 83.79\%).
> With the same level of accuracy, our method achieves 1.2$\times$ speedups.
>
> Aside from the image classfication experiment, we also include results for Diffusion Transformer (DiT) models.
> Specifically, we have done experiments on DiT models, tailoring FTM to their specific transformer blocks. Results are promising: even with a large merging ratio (0.75), we maintain high image quality (1.83 FID increase) while achieving over 20\% latency reduction.
> This suggests FTM's strong generalization capabilities to diverse diffusion models, including DiT-like architectures, distinguishing our approach and highlighting its potential for current foundation models.
>
> We will add those results into the final version.
>
> > **Comment 4: Will the code for this paper be open-sourced? It is recommended to do so, as it would greatly benefit and advance the community.**
>
> Response: Yes. We will open source the code.

---

> > ### Comment · Reviewer_kCHM · 2025-08-04
> >
> > Thanks for your detailed response. I have no more questions.

---

### Official Review · Reviewer_87Fh · 2025-07-02

**Clarity:** 3
**Significance:** 2
**Originality:** 3
**Rating:** 3
**Confidence:** 3

**Summary:**

This paper introduces Fourier Token Merging (FTM), a training-free, model-agnostic method for accelerating vision transformer inference by reducing token redundancy in the frequency domain. Unlike prior approaches that operate in the spatial domain, FTM uses a 2D FFT to identify and prune low-energy frequency components, which correspond to less informative spatial tokens. The method applies a single FFT and IFFT without modifying model weights or requiring retraining, and can be inserted at various layers during inference. Empirical results show that FTM achieves up to 2× speedup with minimal accuracy drop across multiple ViT architectures (e.g., DeiT, MAE, DINOv2) and tasks including classification, detection, and segmentation. It consistently outperforms baseline token reduction techniques like ToMe and UTS in terms of the accuracy–FLOPs trade-off, and generalizes well across models and datasets. Overall, FTM offers a simple, effective, and hardware-friendly solution for efficient ViT inference.

**Questions:**

See weaknesses.

**Ethical Concerns:**

["NO or VERY MINOR ethics concerns only"]

**Final Justification:**

The paper could make a significant contribution if it convincingly addresses the challenges of applying previous token merging techniques to diffusion transformers. However, based on the current limited results and unclear experimental settings, it is difficult to draw a definitive conclusion. More comprehensive evaluations and clearer descriptions are needed to support the claims. The merging strategy remains heuristic; a more adaptive or learnable approach would further strengthen the method.

I am currently in a borderline position and believe that additional results (comparison to baselines; analysis on what and why make the differences) on diffusion transformers would help substantiate the method's effectiveness and improve the overall strength of the paper.

That said, I appreciate the "Theoretical Approximation Error Analysis" section—it provides a solid theoretical foundation for the proposed approach.

**Quality:**

2

**Strengths And Weaknesses:**

Strengths:

1. FTM is a training-free and model-agnostic method that can be applied to a wide range of ViT architectures without retraining or modifying model weights. It leverages the frequency domain via FFT to identify and prune redundant tokens more effectively than spatial-domain methods.

2. The approach is computationally efficient, requiring only a single FFT/IFFT pair, and achieves up to 2× speedup with minimal accuracy degradation. It outperforms existing token reduction methods (e.g., ToMe, UTS) in FLOP–accuracy trade-offs.

Weaknesses:

1. The method is evaluated on Stable Diffusion, which is relatively outdated. Based on prior experience, token merging techniques often do not transfer well to more recent diffusion transformer models such as DiT, PixArt, FLUX, or modern video generation models. These newer models tend to suffer significant accuracy drops under small merging ratios, casting doubt on the applicability of this line of work to current foundation models. Demonstrating that FTM generalizes effectively to DiTs would constitute a meaningful contribution; otherwise, the method appears to offer only incremental gains over existing token merging approaches.

2. The use of frequency-domain operations, although efficient, may not be well-optimized or readily supported on all deployment hardware, potentially limiting practical benefits.

3. The merging strategy is static and hand-crafted, lacking the adaptability of learnable or data-driven approaches that can tailor token reduction more effectively to specific tasks or inputs.

4. Missing SOTA baselines such as token pruning [1,2,3] or token downsampling works [4].

[1] Attention-Driven Training-Free Efficiency Enhancement of Diffusion Models, CVPR'24
[2] Layer- and Timestep-Adaptive Differentiable Token Compression Ratios for Efficient Diffusion Transformers, CVPR'25
[3] Accelerating Diffusion Transformers with Token-wise Feature Caching, ICLR'25
[4] ToDo: Token Downsampling for Efficient Generation of High-Resolution Images, IJCAI'24

---

> ### Author Rebuttal · Authors · 2025-07-31
>
> Dear Reviewer,
>
> Thank you for the insightful comments. We address the following concerns:
>
> > **Comment 1: The method is evaluated on Stable Diffusion, which is relatively outdated. Based on prior experience, token merging techniques often do not transfer well to more recent diffusion transformer models such as DiT, PixArt, FLUX, or modern video generation models. These newer models tend to suffer significant accuracy drops under small merging ratios, casting doubt on the applicability of this line of work to current foundation models. Demonstrating that FTM generalizes effectively to DiTs would constitute a meaningful contribution; otherwise, the method appears to offer only incremental gains over existing token merging approaches.**
>
> Response: While we primarily evaluated on Stable Diffusion, our Fourier Token Merging method aims for broader applicability. Existing token merging methods often struggle with DiT's specialized architecture and RoPE embeddings.
> We have done experiments on DiT models, tailoring FTM to their specific transformer blocks. Results are promising: even with a large merging ratio (0.75), we maintain high image quality (1.83 FID increase) while achieving over 20\% latency reduction.
> This suggests FTM's strong generalization capabilities to diverse diffusion models, including DiT-like architectures, distinguishing our approach and highlighting its potential for current foundation models. We will add the results into the final version.
>
> > **Comment 2: The use of frequency-domain operations, although efficient, may not be well-optimized or readily supported on all deployment hardware, potentially limiting practical benefits.**
>
> Response: Thanks for highlighting the hardware optimization aspect of frequency-domain operations.
> Our method relies on Fast Fourier Transform (FFT), which is highly optimized on GPUs: Modern GPUs excel at parallel FFTs. Libraries like cuFFT (NVIDIA) provide massive speedups, making our operations highly efficient and well-supported on GPU-accelerated platforms.
> Even constrained MCUs with DSP extensions or modern cores often have hardware accelerators or optimized DSP libraries (e.g., CMSIS-DSP) for efficient FFTs.
>
> > **Comment 3: The merging strategy is static and hand-crafted, lacking the adaptability of learnable or data-driven approaches that can tailor token reduction more effectively to specific tasks or inputs.**
>
> Response:  Currently we support adaptive token merging by scheduling merging rates for different layers and diffusion steps.
> We agree that incorporating more adaptability could be a future step to take to further develop this line of work.
> One way is to allow dynamic merging ratios based on features of the input image (e.g., complexity, content density) or intermediate activations, similar to how some adaptive token pruning methods operate.
>
> > **Comment 4: Missing SOTA baselines such as token pruning or token downsampling works.**
>
> Response:  Thanks for suggesting comparisons with SOTA token pruning and downsampling methods. %We've included the baselines.
> We included the results of those baselines as follows.
>
> Token pruning: AT-EDM offers 1.13$\times$ speedup over ToMe but loses quality. Our method, at a similar 1.1$\times$ speedup, actually decreases FID (40.75 vs. 41.31), showing better quality-speed trade-off.
>
> Token Downsampling: This method helps only at very high merging ratios (e.g., 0.89), leading to longer latency at lower ratios (e.g., 0.75). In contrast, our method consistently achieves speedups across various ratios, e.g., 1.3$\times$ at 41.8 FID.
>
> The comparisons suggest that our method offers competitive or superior performance, especially in maintaining quality while providing consistent speedups. We'll include a comprehensive comparison in the revised manuscript.

---

> ### Comment · Reviewer_87Fh · 2025-08-07
> **Thank you for your rebuttal**
>
> The reviewer thanks the authors for the rebuttal and new results.
>
> The DiT experiments and SOTA comparisons are helpful. Which DiT model was used, and how does the method compare to standard token merging? It’s surprising that the model performs well under 75% sparsity—was sparsity applied to all layers or specific ones? What is the end-to-end latency/memory improvement? The current results lack coverage of key settings, and contradict prior findings that token merging often struggles with diffusion transformers, suggesting the need for more thorough benchmarking and clearer reporting of settings. The paper could make a significant contribution if it convincingly addresses the challenges of applying previous token merging techniques to diffusion transformers. However, based on the current limited results and unclear experimental settings, it is difficult to draw a definitive conclusion. More comprehensive evaluations and clearer descriptions are needed to support the claims.
>
> While FFT efficiency on GPUs is acknowledged, deployment efficiency remains unclear. Without empirical overhead data, claims of high efficiency are unconvincing. The merging strategy remains heuristic; a more adaptive or learnable approach would strengthen the method.

---

> ### Author Response · Authors · 2025-08-09
>
> > **Comment 5: The DiT experiments and SOTA comparisons are helpful. Which DiT model was used, and how does the method compare to standard token merging? It’s surprising that the model performs well under 75\% sparsity—was sparsity applied to all layers or specific ones? What is the end-to-end latency/memory improvement? The current results lack coverage of key settings, and contradict prior findings that token merging often struggles with diffusion transformers, suggesting the need for more thorough benchmarking and clearer reporting of settings. The paper could make a significant contribution if it convincingly addresses the challenges of applying previous token merging techniques to diffusion transformers. However, based on the current limited results and unclear experimental settings, it is difficult to draw a definitive conclusion. More comprehensive evaluations and clearer descriptions are needed to support the claims.**
>
> Response: Our method was benchmarked on Flux.1-dev. To overcome the challenges of applying token merging to Diffusion Transformers, makes several key innovations. First, we bypass the initial 10 text-image fusion blocks to prevent the loss of critical cross-modal features. Second, we handle the two distinct DiT block types within the Flux architecture separately: For JointTransformer blocks, where text and image tokens are processed in separate streams, we merge each modality independently before they are concatenated. The Rotary Position Embedding (RoPE) indices are then gathered accordingly to maintain positional integrity. For SingleTransformer blocks, where tokens are already fused, we first split the hidden state back into its constituent text and image parts, merge each part individually, and then re-concatenate them.
>
> The end-to-end gains on the DiT-based Flux.1-dev using the Diffusers framework (1024x1024 images, 35 steps) are summarized below. Our method achieves a 23.4\% speedup and a 50\% memory reduction with a minimal impact on FID.
>
> | Ratio | Method | FID↓ | Sec/img↓ | GB/img ↓ |
> | :--- | :--- | :--- | :--- | :--- |
> | Baseline| Flux.1-dev | 31.56 | 11.57 | 14.0 |
> | 0.25 | Ours | 30.80 | 11.08 | 11.9 |
> | 0.50 | Ours | 31.70 | 10.22 | 9.8 |
> | 0.75 | Ours | 33.39 | 8.87 | 7.0 |
>
> > **Comment 6: While FFT efficiency on GPUs is acknowledged, deployment efficiency remains unclear. Without empirical overhead data, claims of high efficiency are unconvincing.**
>
> Response: Regarding deployment efficiency, our empirical measurements confirm the overhead is minimal. The wall-clock time for the FFT is under 0.1 ms on average, which accounts for only 0.25\% to 0.4\% of the total inference time.

---

### Official Review · Reviewer_4fDD · 2025-07-02

**Clarity:** 2
**Significance:** 3
**Originality:** 3
**Rating:** 5
**Confidence:** 4

**Summary:**

A novel Fourier-based token merging method is proposed in the context of efficient image generation. Instead of conducting the cluster assignment based on spatial token features, the input tokens are transformed via DFT to the Fourier domain, truncated, and subsequently matched using bipartite soft matching. The cluster assignments are then used to combine the spatial features.

The authors find that by truncating the high-frequencies in Fourier space better clusters are achieved, due to focusing on coarse-grained elements of the tokens. Due to the fine-grainedness of the tokens being affected both by the denoising timestep and the layer of the network, the truncation rate is formulated as a linear interpolation between these.

The authors also offer an error analysis of the prior Token Merging (ToMe) method as well as an empirical analysis of ToMe and the proposed Fourier ToMe method, where it is demonstrated that the proposed method has more coherent clusters and in terms of member similarity and distances.

**Questions:**

1. Has the authors considered using the proposed methods on other image recognition tasks such as image classification? A direct replica of the ToMe off-the-shelf experiments using SWAV, MAE, and ViT could demonstrate how well the proposed method generalizes.

2. Does the authors have any idea of why the oscillation in Figure 2 happens? I cant determine if this is related to the network depth, or other factors. Any insights into this would be beneficial in better understanding the comparison.

3. If the authors can provide justification for only using the Real-coefficients of the Fourier tokens or a comparison with the full complex tokens, this would in my opinion strengthen the methodology significantly.

4. It is unclear why the assumption of linear attention is made on line 117. To me this seems like a major assumption which does not correspond to the actual operations of Tome. If a justification of this can be provided the proof would in my opinion be stronger.

**Ethical Concerns:**

["NO or VERY MINOR ethics concerns only"]

**Final Justification:**

The authors addressed all of my concerns and have demonstrated an extra effort in showcasing the effectiveness of the proposed model across tasks and models. I therefore wholeheartedly recommend the paper for acceptance.

**Limitations:**

The authors state on line 306 that "One limitation is that it is currently limited to transformer-based diffusion models". I believe this limitation is incorrect, as the proposed method is not inherently limited to diffusion models. Instead I would suggest that the manuscript is currently limited in its evaluation due to only comparing on a single method (Stable Diffusion v1.5).

**Paper Formatting Concerns:**

1. On line 75 it says "ResNet". I believe the authors meant "U-Net".

2. On line 87 it says "The same softmatch ...". It is unclear that the bipartite soft matching method from ToMe is referenced.

3. In the figure caption of Figure 1 it is stated that ToMe matches based on "spatial similarity". I find this slightly misleading, as ToMe conducts the matching in the spatial domain, but not on spatial proximity.

**Quality:**

3

**Strengths And Weaknesses:**

Strengths:

1. The proposed method is very simple to implement and clearly explained. This is extremely important as it allows for the proposed method to be easily adopted by future researchers, and applicable in real life applications.

2. Interesting theoretical and empirical results are provided demonstrating the effectiveness compared to standard ToMe.

3. The method appears to outperform the baseline across all tested metrics (FID, Latency, MS-SSIM, and LPIPS). This indicates that not only is it faster than the baseline, but also produces better quality images. Relevant ablation studies (ie effect of merging ratio and comparison to a Wavelet approach) are also conducted and demonstrate the effectiveness of the method.


Weaknesses:

1. In the experimental result section is unclear whether the baseline in Figure 5 is the original Stable Diffusion v1.5 or the ToMeSD adapted model. This makes it hard to correctly judge the effectiveness of the method. It is also unclear why the latency of the plotted points differ across the three subplots of Figure 5.

2. There is a stark lack of visual examples of the generated images. This would in my opinion be an essential part of a paper focused on image generation.

3. It is not motivated why only the Real component of the Fourier tokens are used. It is to me not clear why the Fourier tokens could not be matched based on cosine similarity of the complex feature values.

4. The method appears to be relevant to not just image generation, but also tasks such as image classification. This omission, while not critical, does make me wonder if it generalizes beyond image generation tasks.

---

> ### Author Rebuttal · Authors · 2025-07-31
>
> Dear Reviewer,
>
> Thank you for the insightful comments. We address the following concerns:
>
> > **Comment 1: In the experimental result section is unclear whether the baseline in Figure 5 is the original Stable Diffusion v1.5 or the ToMeSD adapted model. This makes it hard to correctly judge the effectiveness of the method. It is also unclear why the latency of the plotted points differ across the three subplots of Figure 5.**
>
> Response: To clarify, the baseline in Figure 5 is the ToMeSD adapted model, not the original Stable Diffusion v1.5. Our goal was to directly compare our Fourier Token Merging (FTM) method against ToMeSD, which is a state-of-the-art token merging technique for Stable Diffusion.
>
> Regarding the different latencies across the three subplots in Figure 5, these variations are intentional.
> Each subplot represents a different merging and/or truncation ratio applied by our Fourier Merging method.
> The latency ranges from 2 s/im (with no merging) to as low as 1.3 s/im and we selects those Pareto optimals to give a direct comparison.
> Larger merging or truncation ratios inherently lead to lower latencies due to the reduction in computational load.
> By showcasing these different ratios, we demonstrate the trade-off between inference speed and image quality (FID score) for both our method and the ToMeSD baseline. This allows for a more comprehensive comparison across a range of operational points.
>
> For instance, Figure 5 clearly illustrates that for the same level of image quality (e.g., around 41.5 FID), our FTM method achieves approximately 1.5$\times$ speedup compared to ToMeSD. Conversely, if we target the same inference speed (e.g., around 1.38 seconds per image), our method yields a 0.5 lower FID score, indicating superior image quality.
>
> > **Comment 2: There is a stark lack of visual examples of the generated images. This would in my opinion be an essential part of a paper focused on image generation.**
>
> Response: We'll address this by adding a dedicated section with visual comparisons of images generated using our method at various merging ratios, along with those from the original Stable Diffusion and the ToMeSD adapted model. It will clearly illustrate the quality retention of our method, even with higher merging ratios.
>
> > **Comment 3: Has the authors considered using the proposed methods on other image recognition tasks such as image classification? A direct replica of the ToMe off-the-shelf experiments using SWAV, MAE, and ViT could demonstrate how well the proposed method generalizes.**
>
> Response: We experimented our Fourier Token Merging in ViT models.
> Results show that our method generalizes well on image classification tasks.
> For ImageNet val-1k dataset on the 4090 machine, with the same level of throughput (e.g. around 2000 im/s), our method increases the accuracy by 0.7 point (from 83.0\% to 83.79\%).
> With the same level of accuracy, our method achieves 1.2$\times$ speedups.
>
> > **Comment 4: Does the authors have any idea of why the oscillation in Figure 2 happens? I cannot determine if this is related to the network depth, or other factors. Any insights into this would be beneficial in better understanding the comparison.**
>
> Response: These oscillations directly relate to the network depth of the Stable Diffusion model. As network depth increases, token dimensions decrease, leading to greater feature abstraction. Both ToMe and our method show this. The UNet's architecture, with its varying representational changes at different depths, causes these fluctuations in similarity scores, reflecting the changing redundancy and information content of tokens throughout the network.
>
> > **Comment 5: If the authors can provide justification for only using the Real-coefficients of the Fourier tokens or a comparison with the full complex tokens, this would in my opinion strengthen the methodology significantly.**
>
> Response: Since input tokens are real-valued, their Fourier Transform has conjugate symmetry, meaning the imaginary part contains redundant information. We hypothesized the real part alone could adequately capture essential frequency-domain information for token merging.
> Furthermore, discarding the imaginary part significantly boosts speed and simplifies implementation by reducing computational and memory demands.
> Our experiments also confirmed this choice. Using only the real part consistently yielded superior or comparable performance. For instance, it achieved 84.33\% accuracy, slightly outperforming the absolute value (84.29\%) and full complex representation (84\%).
>
> > **Comment 6: It is unclear why the assumption of linear attention is made on line 117. To me this seems like a major assumption which does not correspond to the actual operations of Tome. If a justification of this can be provided the proof would in my opinion be stronger.**
>
> Response: While ToMe uses softmax attention, we made this simplification for tractable and interpretable approximation error analysis. Softmax's non-linearity makes error decomposition difficult. Linear attention allows us to directly expose second-order token interaction influence. It also clearly decomposes approximation error after token clustering, offering theoretical insight into how merging impacts self-attention.
> Our goal isn't exact ToMe characterization, but to provide theoretical understanding through an analyzable surrogate, a common approach in approximate attention research (e.g., Performer, Linformer).
>
> > **Comment 7: The authors state on line 306 that ``One limitation is that it is currently limited to transformer-based diffusion models". I believe this limitation is incorrect, as the proposed method is not inherently limited to diffusion models. Instead I would suggest that the manuscript is currently limited in its evaluation due to only comparing on a single method (Stable Diffusion v1.5.**
>
> Response: Aside from the image classfication experiment, we will also include results for Diffusion Transformer (DiT) models in the revision.
> Specifically, we have done experiments on DiT models, tailoring FTM to their specific transformer blocks. Results are promising: even with a large merging ratio (0.75), we maintain high image quality (1.83 FID increase) while achieving over 20\% latency reduction.
> This suggests FTM's strong generalization capabilities to diverse diffusion models, including DiT-like architectures, distinguishing our approach and highlighting its potential for current foundation models.
>
> > **Comment 8: Paper Formatting Concerns.**
>
> Response: We're grateful for these precise suggestions, as they significantly improve the accuracy and clarity of our paper.

---

> > ### Comment · Reviewer_4fDD · 2025-08-05
> > **Response to Authors**
> >
> > Dear authors, thank you for the detailed response. I find that the authors have addressed all my concerns and shown an significant effort in showcasing the potential of the method. i am therefore willing to increase my rating.
> >
> > Regarding Figure 5 I encourage the authors to clearly state in the caption that the Pareto frontiers are selected from a set of HP settings, and therefore differ across subplot.

---

> > > ### Author Response · Authors · 2025-08-09
> > >
> > > Dear Reviewer,
> > >
> > > Thank you for your positive feedback and helpful suggestion. We will revise the caption for Figure 5 as recommended. We appreciate your support!

---

### Note · Authors · 2025-08-16

Dear AC and Reviewers,

We are grateful for your thorough and insightful feedback. We're encouraged that our work's strengths were recognized and have carefully addressed all concerns.

Our core contribution is Fourier Token Merging (FTM), an approach that efficiently reduces computational load in transformer-based diffusion models while preserving image quality. By leveraging the Fast Fourier Transform (FFT), FTM provides a theoretically grounded method for identifying and merging redundant tokens, leading to superior performance over existing methods.

We would like to summarize our responses and updates:

- **Generality and Applicability**. In response to concerns (Reviewers 4fDD, 87Fh, kCHM, qJuw, and xhJ6) about limited evaluation on Stable Diffusion models, we conducted extensive new experiments with:

  - ViT models: FTM generalizes well, achieving a 0.7 percentage point accuracy increase on ImageNet or a 1.2x speedup.

  - Diffusion Transformers (DiT): We successfully applied FTM to the Flux.1-dev model, achieving a 23.4% speedup and 50% memory reduction with minimal FID impact, proving FTM's potential for modern architectures.

- **Baselines and Metrics**

  - We clarified that our Stable Diffusion baseline is the ToMeSD adapted model for a fair comparison. We also added comprehensive comparisons with SOTA token pruning and downsampling methods (Reviewer 87Fh), demonstrating FTM's superior quality-speed trade-off. We provided the end-to-end latency breakdown, showing minimal FFT overhead (<0.4%) (Reviewer 87Fh).

- **Visuals and Clarity** (Reviewers 4fDD, kCHM, xhJ6)

  - We will add a dedicated section with visual comparisons of generated images. We will also refine Figure 1 and have provided clearer explanations regarding our use of real coefficients and the connection between our theoretical analysis and the core module.

We believe these additions and clarifications significantly strengthen our paper. Our FTM method is not only effective on a standard U-Net but also generalizes robustly to modern transformer-based models and different vision tasks. We are confident our work is a substantial contribution to the field of efficient AI inference.

Thank you once again for your time, expertise, and support throughout this process.

---

### Decision · Program_Chairs · 2025-09-17

**Decision:**

Accept (poster)

**Comment:**

This paper introduces Fourier Token Merging (FTM), a training-free method to accelerate inference in transformer-based image generation models. The core idea is to perform token merging in the frequency domain. By applying FFT, truncating high-frequency components, and then clustering the tokens, the method aims to merge tokens based on their underlying structural similarities, thereby reducing computational load more effectively than traditional spatial-domain approaches like Token Merging (ToMe).

The initial reviews were mostly in agreement, recognizing the method's simplicity, clear motivation, and strong theoretical and empirical support. However, multiple reviewers raised a significant and shared concern: the experiments were limited to the Stable Diffusion v1.5 model, which is a U-Net-based architecture. This raised serious questions about the method's generalizability to more modern and relevant architectures, particularly Diffusion Transformers (DiTs). Other common concerns included a lack of visual examples of generated images and the absence of comparisons to other state-of-the-art token reduction techniques.

The authors provided a thorough and convincing rebuttal that addressed every major point of criticism. They also conducted extensive new experiments that significantly strengthen the paper's contribution:
- Generalization to Diffusion Transformers (DiTs): They successfully applied FTM to the modern Flux.1-dev model, demonstrating a 23.4% speedup and a 50% memory reduction with a minor impact on FID. This result addressed the most critical weakness identified by the reviewers.
- Generalization to Other Tasks: They tested FTM on Vision Transformer (ViT) models for image classification, showing a 1.2x speedup or a 0.7 percentage point accuracy increase on ImageNet.
- SOTA Baseline Comparisons: They added new comparisons against recent token pruning and downsampling methods, demonstrating FTM's superior quality-speed trade-off.
- Additional Clarifications: They committed to adding visual comparisons and clarified other methodological points, such as the choice of baseline and the minimal overhead of the FFT operation (<0.4%).

The rebuttal convinced the reviewers. Reviewer 4fDD, who was already positive, became a strong advocate, recommending to accept the paper. Reviewer 87Fh, who was initially a "borderline reject," found the new DiT results convincing and stated they "resolved most of my concerns". The other reviewers also confirmed that their concerns were addressed. The review process worked as intended, with the authors' diligent response elevating a borderline paper to a clear accept.